# ‘*Candidatus* Phytoplasma ziziphi’ Changes the Metabolite Composition of Jujube Tree Leaves and Affects the Feeding Behavior of Its Insect Vector *Hishimonus hamatus* Kuoh

**DOI:** 10.3390/insects14090750

**Published:** 2023-09-06

**Authors:** Rui-Chang Liu, Bo-Liao Li, Xiu-Lin Chen, Jing-Jing Liu, Kun Luo, Guang-Wei Li

**Affiliations:** Shaanxi Province Key Laboratory of Jujube, College of Life Science, Yan’an University, Yan’an 716000, China

**Keywords:** leafhopper, phytoplasma, EPG, metabolome, plant–insect interaction

## Abstract

**Simple Summary:**

Phytoplasma are wall-less phytopathogens that invade hundreds of plant species, causing symptoms, including witches’ broom, phyllody, and leaf yellowing. The difficulty in phytoplasma artificial cultivation makes research on the interactions of phytoplasma–plant–vector insects lag behind the research on interactions of other phytopathogen–plant–vector insects. The spread of phytoplasma heavily relies on piercing–sucking insects. In this study, the feeding behavior of the leafhopper *Hishimonus hamatus* Kuoh fed on healthy Chinese jujube leaves and on jujube witches’ broom (JWB) leaves was investigated to find whether JWB infection changed the feeding behavior or preference of *H. hamatus*. Then, we performed a metabolomic analysis to inspect the metabolite composition of healthy and JWB-infected jujube leaves and tried to explain why the leafhopper tended to feed on JWB-infected leaves. We found that more small-molecular carbohydrates, free amino acids, and free fatty acids and less lignans, coumarins and triterpenoids were accumulated in JWB-infected leaves, which might be related to more vector leafhopper feeding with a higher frequency.

**Abstract:**

*Hishimonus hamatus* Kuoh is a leafhopper species native to China that feeds on Chinese jujube leaves. This leafhopper species has been verified to transmit jujube witches’ broom (JWB) disease, caused by phytoplasma, a fatal plant pathogen, which belongs to the phytoplasma subgroup 16SrV-B. The transmission of JWB phytoplasma largely relies on the feeding behavior of piercing–sucking leafhoppers. However, the specific mechanisms behind how and why the infection of JWB influences the feeding behavior of these leafhoppers are not fully understood. To address this, a study was conducted to compare the feeding patterns of *H. hamatus* when feeding JWB-infested jujube leaves to healthy leaves using the electrical penetration graph (EPG) technique. Then, a widely targeted metabolome analysis was performed to identify differences in the metabolite composition of JWB-infected jujube leaves and that of healthy jujube leaves. The results of EPG analyses revealed that when feeding on JWB-infected jujube leaves, *H. hamatus* exhibited an increased frequency of phloem ingestion and spent longer in the phloem feeding phase compared to when feeding on healthy leaves. In addition, the results of metabolomic analyses showed that JWB-infected leaves accumulated higher levels of small-molecular carbohydrates, free amino acids, and free fatty acids, as well as lower levels of lignans, coumarins and triterpenoids compared to healthy leaves. The above results indicated that the *H. hamatus* preferentially fed on the phloem of infected leaves, which seems to be linked to the transmission of the JWB phytoplasma. The results of metabolomic analyses partially imply that the chemical compounds might play a role in making the infected leaves more attractive to *H. hamatus* for feeding.

## 1. Introduction

Phytoplasma to Mollicutes are prokaryotic plant pathogens, sharing similar morphology and ultrastructure with mycoplasma [1]. They obligately inhabit the phloem of plants and insects, and they are responsible for invading hundreds of plant species worldwide [2,3]. The common symptoms caused by phytoplasma include witches’ brooms, phyllody, leaf yellowing, reduced growth, and so on [4,5]. Phytoplasmas release effectors, which affect the metabolic and defense systems of host plants, thereby altering the basic development processes of plants [6,7].

The jujube (*Ziziphus jujuba* Mill.) is an anciently cultivated fruit tree in China [8]. According to a survey of unearthed kernels, leaf fossils, and traditional Chinese references, together with large-scale population genomic analyses, the side area of the Shanxi–Shaanxi Gorge of the Yellow River is considered the earliest cultivation center [8,9]. Given the rich nutritional and medical value of jujube fruits and the adaptivity of jujube trees to tolerate arid and barren soil [10,11,12,13], the jujube cultivation area spreads worldwide [8,10].

However, the jujube in China suffers from jujube witches’ brooms (JWB) disease, showing symptoms, like witches’ broom, phyllody, leaf yellowing, and weak trees [14]. The infected trees die within several years [15]. JWB is caused by the infection of JWB phytoplasma ‘*Candidatus* Phytoplasma ziziphi’, which belongs to the 16 SrV group (the elm yellow group), 16SrV-B subgroup [16]. Some effectors released by ‘*Ca.* P. ziziphi’ have been identified and verified to target transcriptional factors that affect the normal development of jujube [17,18,19]. For example, JWB phytoplasma effectors SJP1 and SJP2 stimulate the lateral bud outgrowth [18], JWB phytoplasma effector Zaofeng6 induces shoot proliferation [19], and JWB phytoplasma effector SJP3 induces phyllody [17].

Phytoplasmas are transmitted by phloem-feeding insects, including leafhoppers, planthoppers, and psyllids [20,21]. For instance, flavescence dorée disease of grapevine (FD), caused by FD phytoplasma that belongs to the 16SrV-C or 16SrV-D subgroup, is transmitted by the American grapevine leafhopper, *Scaphoideus titanus* Ball [22,23,24]. Likewise, the wheat blue dwarf (WBD) phytoplasma, ‘*Ca.* P. tritici’, which belongs to 16SrI group, is transmitted by the leafhopper *Psammotettix striatus* L. [25]. Previous research has confirmed that *Hishimonoides chinensis* Anufriev [26], *H. aurifascialis* Kuoh [27], *Hishimonus lamellatus* Cai et Kuoh [28,29], and *H. sellatus* [30] could transmit JWB phytoplasma. The above four leafhopper species were found in jujube orchards within or on the edge of the North China Plain. Our group confirmed that *Hishimonus hamatus* Kuoh was the major vector that transmitted JWB phytoplasma on the west side of the Shanxi-Shaanxi Gorge jujube orchards via field trapping and the indoor induction of phytoplasma-carrying leafhoppers to the phytoplasma-free jujube plants [31].

Understanding the probing and feeding behavior of *H. hamatus* on both JWB-infected and non-infected jujube plants is crucial for unraveling the epidemiology of JWB. The use of electrical penetration graphs (EPGs) provided real-time insights into hemipteran feeding behaviors [32,33,34,35]. Therefore, in this study, 8 h recording and analysis of *H. hamatus* probing and feeding behavior on ‘*Ca.* P. ziziphi’ infected and non-infected jujube plants using EPG were conducted. The objective was to explore if the infection of ‘*Ca.* P. ziziphi’ influences the feeding behavior of *H. hamatus* and to understand the potential reasons underlying the feeding preferences in the context of the intricate interactions between insects, phytoplasmas, and plants. To gain a comprehensive understanding, a widely targeted metabolomic analysis based on ultra-high performance liquid chromatography (UPLC-MS/MS) was employed to examine the metabolite composition of JWB-infected and non-infected jujube leaves.

## 2. Materials and Methods

### 2.1. H. hamatus Collection and Rearing

To establish the laboratory population of *H. hamatus*, the adults and nymphs were collected through net capture from JWB-infected jujube orchards in Yanchuan, located on the west side of Shanxi-Shaanxi Gorge, from July to August of 2022. Then, the collected *H. hamatus* were transferred to healthy jujube plants (height ≈ 50 cm) that were placed in nylon cages (60 × 45 × 100 cm) in the laboratory, with each cage accommodating around forty adult leafhoppers. The insects were reared at 23 ± 2 °C with a photoperiod of L:D = 14 h:10 h and with no humidity control. This rearing process occurred from the beginning of September to the middle of October 2022. For the EPG experiments, adults from 7 to 17 days old after emergence were selected, since, in this period, they were sexually mature and actively feeding. Prior to the initiation of EPG experiments, a preliminary assessment was conducted to determine whether the tested leafhopper carried JWB phytoplasma. This assessment involved amplifying a segment of 16S ribosomal DNA (rDNA) of JWB phytoplasma using the polymerase chain reaction (PCR) technique [16].

### 2.2. Plant Rearing

The jujube plants chosen for EPG tests, which were free from JWB phytoplasma, were two-year old seedlings that had been cultivated with a height of about 80 cm through tissue-cultured techniques. These healthy seedlings were cultivated under controlled conditions where the temperature was maintained at 23 ± 2 °C, with a photoperiod of L:D = 14 h:10 h and with no humidity control. Subsequently, approximately one hundred *H. hamatus*, captured from JWB-infected jujube orchards, were introduced to ten healthy seedlings and maintained for two weeks. Both the uninfected healthy plants and those infested with leafhoppers were grown in pots with a capacity of 12.0 L and were watered once a week. The infestation status of jujube plants for electropenetrography or wide metabolomic analysis was determined by amplifying 16S-rDNA from PWB phytoplasma using a nested PCR technique as well as witches’ broom symptoms [16].

### 2.3. Electropenetrography

A Giga-8 DC EPG system was used (W.F. Tjallingii, Dept. of Entomology, Wageningen Agricultural University, Wageningen, The Netherlands) to evaluate the probing activities of *H. hamatus* that feed on JWB-non-infected and JWB-infected jujube leaves. The recordings took place in the Entomology Laboratory at Yan’an University. To ensure consistent conditions, the adults were starved for 4 h before the tests. Five minutes before the tests, the leafhoppers were anesthetized using CO_2_ for two seconds. A gold wire (3 cm length; 20 µm diameter) was attached to the prothorax of each leafhopper using silver glue (EPG Systems, Wageningen, The Netherlands). The other end of this wire was connected to a copper cylinder (2 cm length, 2 mm diameter) that was inserted into the input of the EPG head stage amplifier. Each leafhopper was connected to an amplifier input before being placed on a jujube leaf. Another copper wire electrode (20 cm length, 2 mm diameter) was inserted into the soil of the jujube container. The plants, leafhoppers, and amplifiers were all set up inside a Faraday cage (80 × 60 × 100 cm) to shield the static electricity and other noise sources. The EPG recordings were made in the laboratory at 23 ± 2 °C with a relative humidity of 60 ± 10%. Output was set at 50 × gain, and plant voltage was adjusted so that the EPG signal fit between +5 V and −5 V. All recordings were made between 12:00 and 12:30 every day. A total of 51 recordings were made, and each day, six recordings were run. Each single recording was represented by a different plant–leafhopper combination, one male or one female on JWB-non-infected or JWB-infected jujube. In the case of falling from leaves, the leafhopper was repositioned. At the end of the recording, dead individuals were noted and excluded from further analyses.

EPG signals were acquired and analyzed using Stylet+d software. The EPG waveforms were identified and measured based on the capture signals (as shown in Appendix A): (1) waveform Np, representing non-probing behavior; (2) waveform A, corresponding to stylet pathway phase; (3) waveform C, representing active ingestion of xylem sap; (4) waveform E, representing passive ingestion of phloem sap.

### 2.4. EPG Statistic Analysis

All of the EPG recordings were analyzed using R-based software (v. 4.2.2, R core team). Then, this table was read in the R package *dplyr*, and descriptive statistics were run. In the table of original recording data, each row corresponded to a single recording (a unique combination of one leafhopper and one sex), while each column represented an EPG variable. Univariate analyses were conducted starting from the Generalized Linear Model (GLM): quasi-Poisson for counts and Gamma or inverse-Gaussian for positive continuous variables [33,34]. Pairwise comparison (post hoc) between effects due to JWB infection status and sex was calculated with estimated marginal means and 95% confidence intervals with the function *emmeans* from the “*emmeans*” package and *p*-value adjustment using the method of *Tukey* [35].

A multivariate Canonical Correspondence Analysis (CCA) was performed using *vegan* (v. 2.6-4) [36] and *ggordiplots* (v. 0.4.1) [37]. Four multi-collinear variables were excluded based on a correlation coefficient higher than 0.95 (package *usdm* v. 1.1-18) [38]. The remaining 11 variables were standardized (Hellinger method) and subjected to CCA, with treatment, sex, and their interaction serving as the explanatory variables [33]. The CCA result was confirmed through a permutational Multivariate Analysis of Variance (perMANOVA) [39].

### 2.5. Widely Targeted Metabolome Analysis

The biological samples from Chinese jujube leaves were freeze-dried in a freeze-dryer (Scientz-100F). The dried specimens were homogenized using a mixer mill (MM400, Retch) for 90 s at 30 Hz. Fifty milligrams of lyophilized powder from each sample was dissolved in a 1.2 mL 70% methanol solution. Then, we vortexed 30 s every 30 min six times. After centrifuging at 12,000 rpm for 3 min, the supernatants were filtrated before UPLC-MS/MS analysis. The widely targeted metabolome analysis was carried out on the UPLC-MS/MS system (UPLC: Shim-pack UFLC SHIMADZU CBM30A, Kyoto, Japan; MS/MS: SCIEX QTRAP 6500, Applied Biosystems, Framingham, MA, USA). The analytical conditions were as follows: UPLC: column, Agilent SB-C18 (1.8 µm, 2.1 mm × 100 mm). The mobile phase consisted of solvent A (0.1 formic acid in pure water) and solvent B (0.1% formic acid in acetonitrile). The sample measurements were performed with a gradient program that employed starting conditions of 95% A and 5% B. Within 9 min, a linear gradient of 5% A and 95% B was programmed. The ratio of 95% A and 5% B was kept for 1 min. Subsequently, a composition of 95% A, 5.0% B was adjusted within 1.1 min and kept for 2.9 min. We set the flow velocity at 0.35 mL/min, the column oven at 40 °C, and the injection volume at 4 μL. The primary and secondary spectral properties of metabolites were determined based on public databases of metabolites and the self-constructed MWDB V2.0 database (Metware Biotechnology Co., Ltd. Wuhan, Hubei, China). The metabolites were quantified with triple-quadrupole mass spectrometry.

The raw data signals were analyzed using the supporting software SCIEX (v. 2.0.01) (Framingham, MA, USA). Then, unsupervised principle component analysis (PCA), orthogonal projections to latent structure-discriminant analysis (OPLS-DA), and Pearson coefficients were conducted in R (v. 4.2.2). For two-group analysis, differentially accumulated metabolites (DAMs) were determined by the threshold of VIP ≥ 1 and |(log2(FoldChange)| > 1.

## 3. Results

### 3.1. Feeding Behavior of H. hamatus

The EPG results of *H. hamatus* stylet behavior are listed in Table 1. JWB infection significantly increased the frequency of phloem ingestion (JWB infection: *z* = 0.656, *p* = 3.61 × 10^−3^; interaction of JWB infection and sex: *t* = −0.764, *p* = 0.445), while it did not significantly influence the average duration of phloem ingestion per insect during the total recording period (JWB infection: *t* = −1.928, *p* = 0.066; interaction: *t* = 1.864, *p* = 0.075), although the average duration of phloem salivation for female adults changed from 0.60 min to 24.51 min and for male adults from 12.47 min to 34.43 min. Neither the effects of JWB infection status nor its interaction with sex influenced the frequency of active xylem ingestion (JWB infection: *t* = −0.432, *p* = 0.668; interaction: *t* = 0.284, *p* = 0.778) or the average duration of active xylem ingestion (JWB infection: *t* = −0.903, *p* = 0.371; interaction: *t* = 0.019, *p* = 0.985). There were no significant differences in the durations of pathway waveforms (waveforms A) per female *H. hamatus* between feeding on JWB-infected jujube leaves and on healthy leaves (*t* = 0.332, *p* = 0.988), while there were significant differences in the durations of waveforms A per male between the two treatments (*t* = 3.149, *p* = 0.015). Neither JWB infection (*t* = 0.270, *p* = 0.789) nor the interaction between infection and sex (*t* = −0.725, *p* = 0.472) affected the number of non-probing frequencies. However, for male adults, the average duration of non-probing waveforms was significantly shorter on JWB-infected jujube leaves than on healthy jujube leaves (*t* = −3.083, *p* = 0.017), while for females, the average duration of non-probing waveforms did not show a significant difference between JWB-infected and healthy leaves (*t* = 0.628, *p* = 0.923).

To examine the comprehensive impact of the explanatory variables treatment, sex, and their interactions on *H. hamatus* behavior, a constrained Canonical Correspondence Analysis (CCA) was performed (Figure 1). The non-multi-collinear variables that are more closely associated with the various groups are represented by CCA. In particular, ellipses were drawn containing 99% confidence intervals for standard errors associated with the treatment variable, considering the absence of effects for sex and treat × sex (Table 2). This representation emphasized the distinction between *H. hamatus* feeding on JWB-infected and healthy jujube leaves. The results of the perMANOVA supported the findings of the CCA (Table 2), which highlighted significant differences between JWB-infected and healthy leaves, while neither sex nor the combination of treat and sex showed any significant differences.

### 3.2. Metabolomic Analysis of Ziziphus jujuba Leaves Infected by Phytoplasma

To reveal potential metabolic pathways responding to phytoplasma infection, we used UPLC-MS/MS termed as widely targeted metabolomics to qualitatively and quantitatively detect metabolites in jujube leaves. Firstly, the result of principal component analysis (PCA) suggested that PC1 could annotate 77.3% variety, indicating a clear separation of metabolomics from two groups and the high reliability of this experiment (Figure 2a). In addition, the result of orthogonal projections to latent structure-discriminant analysis (OPLS-DA) indicated a distinct division of metabolites between the phytoplasma-infected group and the healthy group (R^2^X = 0.939, R^2^Y = 1, Q^2^ = 0.999) (Figure 2b). As the Q^2^ value of the OPLS-DA model was larger than 0.9, the evaluation model was stable.

There were 1183 metabolites detected in the samples, including 66 saccharides, 64 free fatty acids, 19 glycerol esters, 29 phosphatidylcholines (LPCs), 23 phosphatidylethanolamine (LPEs), 95 amino acids and derivatives, 19 vitamins, 64 organic acids, 183 phenolic acids and derivatives, 82 alkaloids, 266 flavonoids, 128 terpenoids, 43 lignans and coumarins, 11 tannins, and some other metabolites (Table 3).

We found 646 significantly differentially accumulated metabolites (DAMs) between the JWB-infected group (IN) and the control (CK), with 252 down-regulated and 394 up-regulated metabolites, and the criteria of |log2FoldChange| > 1 and variable importance in project (VIP) > 1 (Table 3 and Appendix A). Among them, 3-hydroxylup-20(29)-en-28-al (Betulinaldehyde) and 3-hydroxylup-12(13),20(29)-diene belonged to triterpenes, with the most down-regulated metabolites in the IN group, with log2FoldChange of −15.96 and −14.96, respectively. Kaempferol-3-O-robinoside-7-O-rhamnoside and Isorhamnetin-3-O-rutinoside-7-O-rhamnoside, which belonged to flavonols, were the most upregulated metabolites in the IN group, with log2FoldChange of 16.39 and 15.32, respectively. In addition, for the primary metabolites, 22.7% saccharides (15 out of 66), 61.2% lipids (85 out of 139), and 62.1% amino acids and derivates (59 out of 95) significantly differed between the two groups. For the secondary metabolites, 46.6% of flavonoids (93/128), 72.7% of terpenoids (93/128), and 56.8% of phenolic acids (104/183) significantly differed between the two groups (Table 3 and Appendix A).

## 4. Discussion

### 4.1. JWB Infection Affected the Feeding Behavior of Hishimonus hamatus

To investigate the potential factors contributing to varying transmission efficiencies of JWB phytoplasma, this study focused on analyzing the feeding behavior of the JWB leafhopper vector *H. hamatus* on both JWB-infected and healthy jujube leaves. This investigation is crucial since phytoplasmas are phloem-limited plant pathogens. Consequently, the phases of phloem feeding are of particular importance as they are closely linked to the vector’s ability to acquire and transmit the phytoplasma pathogen [33].

The EPG technique has been widely employed to study interactions involving plant pathogens, piercing–sucking insect-host plant, and host plants [39,40,41]. In the current study of *H. hamatus* feeding on jujube leaves, the results showed that male *H. hamatus* feeding on JWB-infected leaves exhibited a notably shorter non-probing phase and an extended pathway phase compared to males feeding on healthy leaves. Both sexes performed significantly more phloem probing events and longer phloem ingestion durations on JWB-infected leaves than healthy leaves. This observation corresponds with findings from studies involving other vector insects and their interactions with phytoplasma-infected plants. In the study of the leafhopper *Scaphoideus titanus* Ball that transmits Flavescence dorée, a longer duration of phloem ingestion events was found on the FD-susceptible cultivar Barbera than for the FD-tolerant cultivars Brachetto and Moscato [33]. Similarly, during the EPG analysis of the leafhopper *Matsumuratettix hiroglyphicus* Matsumura, which transmits sugarcane white leaf (SCWL) disease phytoplasma, longer durations of waveform C (phloem salivation) and waveform D (phloem ingestion) were recorded when feeding on symptomatic and asymptomatic SCWL-infected sugarcane plants compared to healthy plants [42]. However, ‘*Ca.* Phytoplasma mali’-infected apple plants did not affect the phloem ingestion behavior of the summer apple psyllid *Cacopsylla picta* Flor [43]. Conversely, infection with ‘*Ca.* P. mali’ did not appear to affect the phloem ingestion behavior of *C. picta*. To gain a deeper understanding of these dynamics, more EPG data focused on phytoplasma–insect vectors are necessary.

### 4.2. JWB Infection Changed the Metabolite Composition of Chinese Jujube Leaves

The extended duration spent in the phloem phase by piercing–sucking insects could, indeed, signify an increased likelihood of transmitting phytoplasma to non-infected plants, given that phytoplasmas are obligate inhabitants of the plant phloem. In the context of persistently transmitted phytoplasma, successful transmission requires insect vectors to transfer the pathogens from infected plants to healthy ones [42,43]. However, the relationship between phytoplasma infection and the survival and development of vector insects is complex. While prolonged phloem feeding might enhance transmission potential, phytoplasma infection might not necessarily facilitate the overall survival and development of vector insects. In fact, interactions between phytoplasma and vector insects can be multifaceted. To shed light on the potential reasons making *H. hamatus* feeding on JWB-infected jujube leaves longer than those on healthy leaves in the phloem ingestion phase, a widely targeted metabolomic analysis was performed to research the differential metabolite composition between JWB-infected jujube leaves and healthy leaves.

#### 4.2.1. Changes of Carbohydrates

The metabolomic analyses revealed the presence of fifteen saccharides in both JWB-infected and healthy jujube leaves. Among them, twelve sugars were up-accumulated, including Glucose-1-phosphate, D-Glucose-6-phosphate, D-Fructose-6-Phosphate, Ribulose-5-phosphate, D-Erythrose-4-phosphate, D-Sedoheptuose-7-phosphate, D-Ribose, D-Glucoronic acid, D-Arabitol, Sorbitol-6-phosphate, D-Glucosamine-1-phosphate, and 1-(sn-Glycero-3-phospho)-1D-myo-inositol (Table 3, Figure 3). Interestingly, the up-accumulation of these carbohydrates could be attributed to the altered metabolism in JWB-infected jujube leaves. Since phytoplasma lacks genes that code the components of certain metabolic pathways, like the tricarboxylic acid cycle and the pentose phosphate pathway [14,44], these carbohydrates may serve as energy sources for phytoplasma and *H. hamatus* or as intermediate products to synthesize other metabolites.

#### 4.2.2. Changes of Lipids

In the present study, the metabolomic analyses revealed that there were increased accumulations of 17 free fatty acids, 11 glycerol esters, 28 lysophosphatidylcholines (lysoPCs), and 21 lysophosphatidyl ethanolamines (lysoPEs), and only 1 decreased free fatty acids, 2 glycerol esters, and 2 lysoPEs in JWB-infected leaves compared to healthy leaves (Figure 3, Appendix A). Similar results were observed between phytoplasma-infected and healthy leaves of sweet cherry leaves [45]. The elevated lipid accumulation might be attributed to the destruction of phospholipid bilayers in the host plant caused by phytoplasma infection. Fatty acids serve as fundamental components for constructing membrane lipids, which are essential throughout the life cycle of any cellular organism [46]. Given that phytoplasma lacks the mechanism for *de novo* fatty acid synthesis, it is plausible that these organisms import the necessary components for membrane lipid assembly from their host plants [14,45,46,47,48]. A lack of unsaturated fatty acids could influence the survival and development of insects. For instance, the loss of the FATTY ACID DESATURASE 7 (FAD7) *Arabidopsis thaliana* L mutation variety impacted the growth of *Myzus persicae* Sulzer [49]. Perhaps the up-accumulation of unsaturated fatty acids in JWB-infected jujube leaves facilitated the survival and development of *H. hamatus*.

#### 4.2.3. Changes of Amino Acids

Amino acids participate in various processes of plant growth, development, and homeostasis [50]. While phytoplasma lacks genes for *de novo* amino acid synthesis, amino acids from the sieve tube elements of host plants might facilitate phytoplasma multiplication [45]. Furthermore, the amino acids within host plants can influence the preferences of sap-feeding insects [51]. If increased levels of amino acids in host plants, caused by phytoplasma infection, benefit the survival and development of sap-feeding phytoplasma vectors, this could lead to a feedback loop promoting the spread of phytoplasma. In the current study, elevated levels of arginine, histidine, valine, leucine, asparagine, tyrosine, tryptophan, and phenylalanine were found in JWB-infected jujube leaves (Table 3, Figure 4). Similar observations were noted in sweet cherry virescence leaves, elevated levels tyrosine, tryptophan, and phenylalanine, which were invaded by phytoplasma that belonged to the 16SrV-B phytoplasma subgroup [45]. Likewise, ‘*Ca.* P. mali’-infected apple leaves showed higher amounts of alanine, serine, aspartic acid, asparagine, and threonine compared to healthy apple leaves [52]. The relationship between phytoplasma infection and amino acid composition necessitates further exploration to understand its implications for phytoplasma–vector interactions.

#### 4.2.4. Changes of Lignans and Coumarins

Lignans are phenylpropanoid dimers where the phenylpropane units are linked by the central carbon (C8) of their side chains [53]. These compounds have shown antifeedant and deterrent activities on various insects, including coleopteran [54], dipteran [55], hemipteran [56], and lepidopteran pests [57]. In the current research, it was observed that sixteen out of twenty-four lignans were down-accumulated in JWB-infected jujube leaves, while only three lignans were up-accumulated (Table 3, Figure 5). The differential accumulation of lignans might be related to the preference of *H. hamatus* to feed on JWB-infected jujube leaves.

Coumarins are phenolic substances composed of fused benzene and α-pyrone rings, derived from the shikimate pathway. The coumarin 2H-1-benzopyran-2-one exhibited high toxicity to *Myzus persicae* Sulzer [58]. Natural coumarin shows toxicity to *Spodoptera litura* Fabricius via the inhibition of detoxification enzymes and glucometabolism [59]. In the present study, a decrease in coumarin levels could potentially be linked to the feeding preference of *H. hamatus* for JWB-infected jujube leaves (Appendix A).

#### 4.2.5. Changes of Triterpenoids

Triterpenoids are one of the most numerous and diverse groups of plant secondary metabolites. They exist in either simple, unmodified form or as conjugates with carbohydrates and other macromolecules, especially as triterpene glycosides [60]. Triterpene glycosides contribute to the resistance of pest insects and pathogens, as a group of them showed antifeedant, growth inhibition, and poison abilities [60,61]. For example, asiatic acid isolated from *Shorea robusta* Sal inhibited the growth and extended the instar duration of *Oxya fuscovittata* Maschall [60]. α- and β-amyrin acetates, extracted from *Manilkara subsericea* Mart., exhibited potent growth inhibitory effects on two phytophagous hemipteran insects, *Dysdercus peruvianus* Guérin-Méneville and *Oncopeltus fasciatus* Dallas [62]. In the present study, the accumulation level of asiatic acid was 6.82% in JWB-infected jujube leaves compared to healthy jujube leaves. Similarly, the accumulation level of β-Amyrone was 25.32% in JWB-infected leaves compared to healthy leaves (Appendix A). The reduction in triterpenoids is hypothesized to influence the feeding preference of *H. hamatus* on JWB-infected jujube leaves. However, further consideration is warranted regarding the impact of specific triterpenoids on the survival and development of this leafhopper species.

The presence of higher quantities of low-molecular carbohydrates, free amino acids, and free fatty acids in JWB-infected leaves could potentially contribute to the leafhopper’s preference to feed on leaves. These compounds may offer enhanced nutritional resources or energy sources for the leafhoppers, thereby influencing their feeding behavior. Conversely, the reduced amounts of lignans, coumarins and triterpenoids in JWB-infected leaves could also play a role in shaping the leafhopper’s feeding preference. These compounds might deter or affect the leafhopper’s perception of the leaves, impacting their feeding decisions.

## 5. Conclusions

The results of this work lead us to the conclusion that the leafhopper *H. hamatus* exhibited distinct feeding behavior when consuming JWB-infected jujube leaves compared to healthy leaves. Specifically, the leafhoppers that fed on JWB-infected leaves displayed increased frequency and a longer duration of phloem ingestion. This altered feeding behavior might be attributed, in part, to the differing metabolite composition between the two types of leaves. This research could contribute to a deeper understanding of plant–phytoplasma–insect interactions and their implications for the spread of phytoplasma-associated diseases.

## Figures and Tables

**Figure 1 insects-14-00750-f001:**
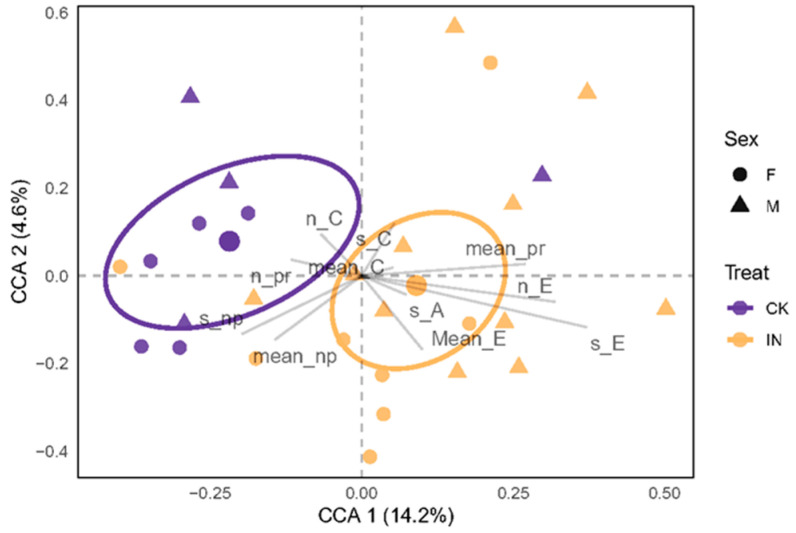
Canonical Correspondence Analysis (CCA) on recordings with phloem phases. The condensed CCA variables explained 13.4% (CCA 1, x axis) and 5.6% (CCA 2, y axis) of the variability. JWB infection recordings were grouped with ellipses, representing 99% confidence intervals for the standard errors, and the centroid of each was represented. Every point represents a single recording: the color denotes the jujube’s level of infection, and the shape denotes the sex of the leafhopper. CK and IN represent the leafhoppers that feed on healthy leaves and JWB-infected leaves, respectively. F and M represent female and male leafhoppers, respectively.

**Figure 2 insects-14-00750-f002:**
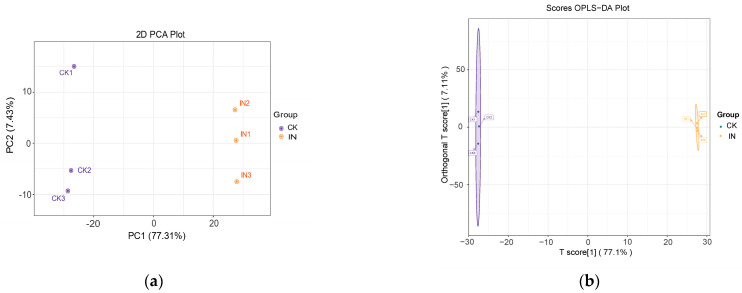
Multivariable analysis of the metabolomic data derived from the metabolites of healthy and JWB phytoplasma infected jujube leaves. (**a**) Principal component analysis (PCA) analysis; (**b**) orthogonal partial least-squares discriminant analysis (OPLS-DA). CK, healthy leaves. IN, JWB-infected leaves.

**Figure 3 insects-14-00750-f003:**
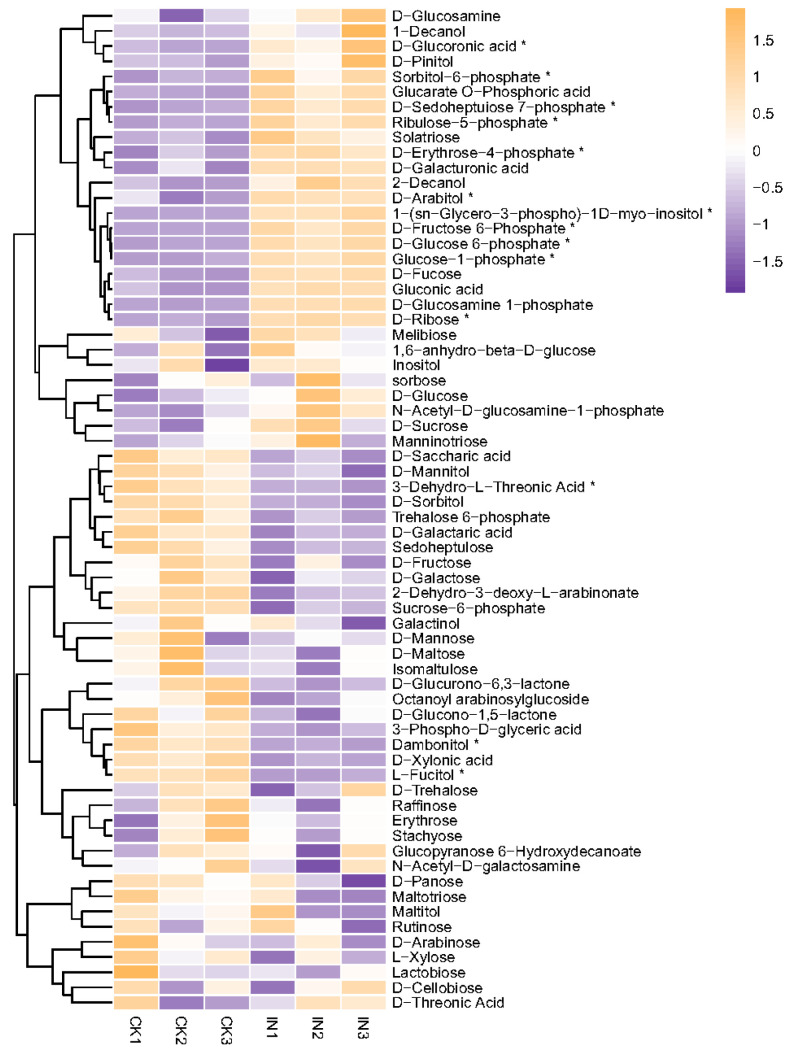
Heatmap of saccharides. The vertical axis shows the clustering of samples; the horizontal shows the sample names. The depth of the purple color represents the relative level of the down-regulated metabolites, and orange represents down-regulated metabolites. The * symbol represents significantly different amounts between healthy and JWB-infected leaves. CK, healthy leaves. IN, JWB-infected leaves.

**Figure 4 insects-14-00750-f004:**
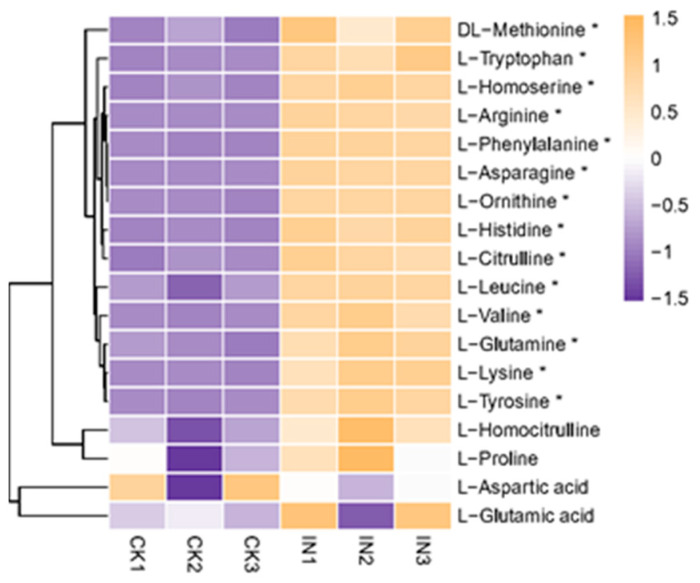
Heatmap of amino acids. The vertical axis shows the clustering of samples; the horizontal shows the sample names. The shorter the cluster branch, the higher the similarity. The depth of the purple color represents the relative level of the down-regulated metabolites, and orange represents down-regulated metabolites. The * symbol represents significantly different amounts between healthy and JWB-infected leaves. CK, healthy leaves. IN, JWB-infected leaves.

**Figure 5 insects-14-00750-f005:**
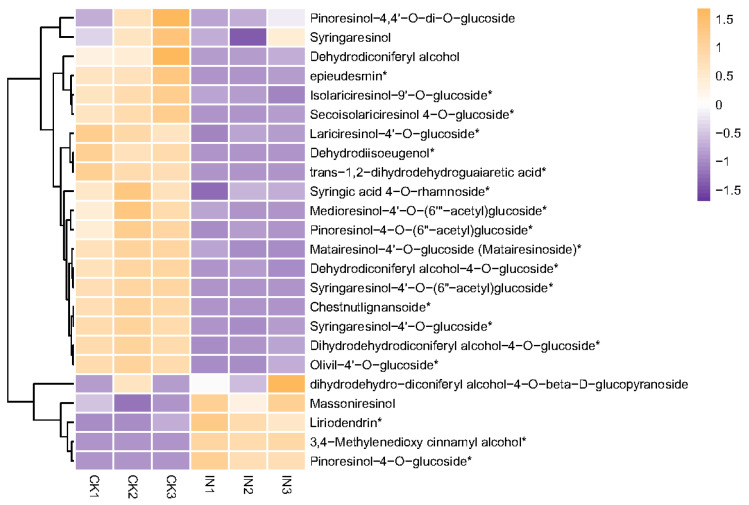
Heatmap of lignans. The vertical axis shows the clustering of samples; the horizontal shows the sample names. The shorter the cluster branch, the higher the similarity. The depth of the purple color represents the relative level of the down-regulated metabolites, and orange represents down-regulated metabolites. The * symbol represents significantly different amounts between healthy and JWB-infected leaves. CK, healthy leaves. IN, JWB-infected leaves.

**Table 1 insects-14-00750-t001:** Median ± SE of variables related to EPG recordings. Each column represents a single combination of whether the jujube plants were infected by JWB and leafhopper sex.

Treat	Healthy		JWB-Infected	
Sex	Female	Male	Female	Male
Number of insects	11	12	12	16
Number of non-probing periods *	154.0 ± 17.8 ^a^	130.0 ± 34.0 ^a^	137.5 ± 29.4 ^a^	123.0 ± 37.5 ^a^
Total duration of non-probing periods [min] **	225.3 ± 21.1 ^a^	266.8 ± 27.6 ^a^	218.0 ± 28.8 ^ab^	113.1 ± 16.8 ^b^
Number of probes *	153.0 ± 17.8 ^a^	129.5 ± 34.1 ^a^	137.0 ± 29.4 ^a^	123.0 ± 37.5 ^a^
Total probing time [min] *	134.7 ± 21.1 ^b^	92.7 ± 27.7 ^b^	142.0 ± 28.8 ^a^	245.8 ± 16.8 ^a^
Total duration of pathway phase [min] **	51.4 ± 18.8 ^b^	22.3 ± 6.5 ^b^	70.9 ± 16.6 ^b^	119.1 ± 19.2 ^a^
Number of active xylem ingestion phases *	106.0 ± 27.2 ^a^	88.5 ± 39.5 ^a^	93.5 ± 21.1 ^a^	100.5 ± 20.7 ^a^
Total duration of active xylem ingestion [min]*	76.0 ± 8.4 ^a^	52.5 ± 25.8 ^a^	73.0 ± 24.2 ^a^	118.0 ± 27.8 ^a^
Mean duration of a single event of pathway phase [min] *	0.8 ± 0.2 ^a^	0.9 ± 0.1 ^a^	0.8 ± 0.2 ^a^	1.1 ± 0.3 ^a^
Number of phloem ingestions *	0 ± 0.72 ^b^	0 ± 5.62 ^b^	8.5 ± 7.8 ^a^	17.5 ± 12.6 ^a^
Total duration of phloem ingestions [min] *	0 ± 0.12 ^b^	0 ± 3.78 ^b^	7.8 ± 6.7 ^a^	3.70 ± 8.67 ^a^
Mean duration of a single event of phloem ingestion [min] *	0.1 ± 0.1 ^a^	0.4 ± 0.1 ^a^	0.8 ± 0.6 ^a^	0.8 ± 0.3 ^a^
Percentage of non-probing time [%] #	52.0 ± 10.1 ^a^	25.4 ± 10.0 ^b^	47.3 ± 8.9 ^a^	20.7 ± 3.9 ^b^
Percentage of probing time spent in pathway-phase [%] #	34.7 ± 5.6 ^a^	60.4 ± 10.8 ^b^	30.7 ± 6.9 ^a^	49.5 ± 6.3 ^b^
Percentage of probing time spent in active xylem ingestion [%] *	13.4 ± 4.7 ^a^	13.8 ± 3.4 ^a^	16.0 ± 3.3 ^a^	20.4 ± 3.8 ^a^
Percentage of probing time spent in passive phloem ingestion [%] *	0.1 ± 0.1 ^b^	0.5 ± 2.7 ^b^	4.1 ± 1.7 ^a^	5.3 ± 2.1 ^a^
Time to 1st probe [min] *	42.0 ± 32.1 ^a^	65.7 ± 13.5 ^a^	67.6 ± 32.1 ^a^	5.0 ± 19.5 ^a^
Time to 1st phloem ingestion [min] *	38.7 ± 40.2 ^a^	92.3 ± 69.9 ^a^	16.9 ± 7.3 ^a^	10.0 ± 32.6 ^a^
Number of probing times before 1st phloem ingestion *	9.0 ± 45.8 ^a^	3.0 ± 85.8 ^a^	3.5 ± 3.8 ^b^	0.0 ± 0.1 ^b^

Each column reports a single combination of Treat and Sex. Each row represents a specific variable: quasi-Poisson or negative-binomial for counts, Gamma or inverse-Gaussian for continuous time variables, and beta-regression for proportions. In the case of no effect for Sex and Treat × Sex, the GLM was run with only Treat as an explanatory variable (indicated in the tables with the * sign after the specific variable name). In the case of effect for Sex or Treat × Sex, GLM was run with all three explanatory variables (indicated in the tables with the ** sign after the specific variable name). In the case of only an effect for Sex, the GLM was run with only Sex as an explanatory variable (indicated in the tables with the # symbol after the specific variable name). Post-hoc comparisons were conducted with the *Tukey* method for *p*-value adjustment at significant levels of 0.05 and 95% confidence intervals. Different letters in each column represented that there is significant difference among different groups after post-hoc comparisons.

**Table 2 insects-14-00750-t002:** perMANOVA results based on Hellinger dissimilarities, using all the non-multi-collinear EPG variables. Df: degrees of freedom; SumOfSqs: sequential sums of squares; F:F statistics values by permutations; Pr(>F): *p*-values, based on 9999 permutations (the lowest possible *p*-value is 0.0001); **: Pr < 0.01 based on the results of perMANOVA.

	Df	Sum of Sqs	R^2^	F	Pr (>F)	Signif
Treat	1	0.1954	0.1243	3.8094	0.0099	**
Sex	1	0.1313	0.0836	2.5608	0.0560	
Treat × Sex	1	0.0136	0.0087	0.2654	0.8751	
Residual	24	1.2308	0.7834	NA	NA	
Total	27	1.5711	1	NA	NA	

**Table 3 insects-14-00750-t003:** A summary of identified and differentially regulated metabolites in jujube leaves responding to JWB infection.

Main Class	Number of Metabolites		
	Identified	Down-Regulated	Up-Regulated
**Alkaloids**	82	13	34
Alkaloids	30	2	9
Aporphine alkaloids	6	0	3
Benzylphenylethylamine	1	0	1
Phenolamine	26	6	15
Piperidine alkaloids	1	0	1
Plumerane alkaloids	13	2	7
Pyridine alkaloids	2	1	0
Quinoline alkaloids	2	1	0
Tropan alkaloids	1	1	0
**Amino acids and derivatives**	95	7	51
**Flavonoids**	266	50	74
Anthocyanidins	10	2	4
Chalcones	15	2	8
Flavanols	15	4	1
Flavanones	29	7	11
Flavanonols	8	3	3
Flavones	68	13	20
Flavonols	114	18	25
Isoflavones	6	1	2
**Lignans and Coumarins**	43	27	4
Coumarins	19	11	1
Lignans	24	16	3
**Lipids**	139	5	80
Free fatty acids	64	1	17
Glycerol ester	19	2	11
Lysophosphatidyl choline	29	0	28
Lysophosphatidyl ethanolamine	23	2	21
Phosphatidyl choline	1	0	1
Spshingolipids	3	0	2
**Nucleotides and derivatives**	52	4	28
**Organic acids**	64	5	24
**Others**	27	10	5
Aldehyde compounds	2	1	0
Chromone	4	1	0
Ketone compounds	6	2	3
Others	15	6	2
**Saccharides**	66	3	12
**Vitamines and derivatives**	19	7	2
**Quinones**	8	1	1
**Tannins**	8	0	1
**Phenolic acids**	183	41	64
**Terpenoids**	128	80	13
Triterpene	96	78	3
Triterpene Saponin	32	2	10

## Data Availability

The data presented in this study are available on request from the corresponding author.

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
