# Peer review of "Candidatus Phytoplasma ziziphi’ Changes the Metabolite Composition of Jujube Tree Leaves and Affects the Feeding Behavior of Its Insect Vector Hishimonus hamatus Kuoh"

_insects, 2023, doi:10.3390/insects14090750_

Round 1

Reviewer 1 Report

The authors provided interesting data that should add more information to plant-insect-phytoplasma interactions knowledge base. However, the manuscript requires major revisions. The authors did not draw accurate conclusions from their data, and this led the tone of the manuscript astray.

As a non-native English speaker, I would strongly suggest using a paid service or find a native English speaker specialist who would correct the grammar, and style mistakes. The quality of presentation suffers from the bad sentence style, wrong word, and term usage, nonuniform term usage, misspelling, some abbreviations, and table contents are not explained.

I recommend providing the insect species authorship for all first mentions of insect species (Hishimonus hamatus Kuoh), as it was done for the plant species.

The authors did not provide the information on data and methods which were used to show that plant and insect material was free from phytoplasma. Also, they did not provide the information how the infected plants were obtained, and how they confirmed that the plants were infected with `Ca. Phytoplasma ziziphi’.

The terms describing feeding behavior should be used uniformly or should be explained very clearly in the text and tables as it produces a lot of confusion. For example, does the term active ingestion (in the table1) corresponds to the term active xylem ingestion (in the text)?

Additionally, according to the table 1. data the insects fed significantly more time on the xylem (I presume that the term active ingestion corresponds to active xylem ingestion) than on the phloem. Why the phloem feeder fed for so long on the xylem? It can raise a discussion that the change of the xylem metabolites affected the feeding behavior.

More emphasis should be put on the fact that the metabolites of the whole leaf were analyzed in this study. This only partially and indirectly can suggest that the arrangement of metabolites in the phloem also changed, and the data do not show the actual composition of the phloem sap in this study. The statement that only the increase of seldom metabolites changed the feeding behavior is inaccurate because the opposite can also be true. I recommend revising the manuscript sections containing such statements and rewrite them accordingly.

I suggest expanding the discussion section with the inclusion of the data on the downregulated metabolites and discuss in what way they can influence the feeding behavior and benefited the insect. The inclusion of such data would increase the value of this manuscript.

Title

"‘Candidatus Phytoplasma ziziphi’ changes the jujube tree phloem composition and affects the feeding behavior of its insect vector Hishimonus hamatus" - The data do not show that phloem composition changed. They only show that the composition of metabolites of the whole leaf changed. I suggest changing the title accordingly.

Simple summary/abstract

“We found that more small-molecular carbohydrates, free amino acids, and free fatty acids were accumulated in JWB-infected leaves, which attracted more vector leafhopper feeding with a higher frequency.”

In my opinion, the observed higher amounts of some metabolites in the whole leaf can only partially prove that this could be the cause of the change in the feeding behavior. The actual change of the phloem metabolites is not reported in this study. It can also be argued that the decrease of some metabolites partaking in the plant defense could produce the same effect. Additionally, the metabolomics data only showed the general trend in the whole leaf. They did not show if the same trend is present in the phloem sap, and what change it inflicted on the compounds in the phloem.

1. Introduction

I recommend including introductory information regarding electropenetrography and its use in the phytoplasma insect-vector research.

2. Materials and Methods

The sticky trap method can only prove the presence of the insects in the orchard. It does not prove phytoplasma infection.

There is no information in this section on what method was used to check for the phytoplasma infection in the plants used for the insect rearing, noninfected/infected plants, and insects. The insects were caught in the orchard so there is a higher probability that some of them could have been infected.

Also, there is no information how the infected plants were acquired, and how it was confirmed that the plants were infected with `Ca. Phytoplasma ziziphi’.

2.4 EPG Statistic Analysis

In this section it should be clearly stated that all statistical analyses were performed using the R based software or it should be pointed out what other software was used in each case.

2.3 Electropenetrography

It is reported that Hishimonus sellatus Uhler species has diurnal changes in stylet insertion behavior. Does H. hamatus possess similar behavior? Did you adjust your experiment regarding this fact? Please comment.

3. Results

3.1. Stylet probing behavior of H. hamatus

The feeding behavior describing terms are not uniform throughout this and Discussion paragraphs which makes it difficult to read and analyze the manuscript. I suggest unifying the terms according to the Table 1.

Please clarify if the term total duration of active ingestion is the same as the term active xylem ingestion. If yes, please use active xylem ingestion in the Table 1.

Please clarify if the terms phloem salivation and phloem phase are the same as the phloem ingestion in Table 1.

Please could you explain why the median values are used in the table? It is known that medians can ignore extreme values, and in that way can skew the view on the distribution of the values.

I suggest providing a supplementary table for the data and values used in the text of this section as they are not included in other tables, and their origin is not explained.

4. Discussion

I suggest moving the tables and figures to the results paragraph.

The phytoplasmas reside in the phloem, and the insects are also phloem feeders. This fact requires that more focus should be given to the composition of the metabolites in this tissue.

It was more discussed what benefit was brought by the increase of some metabolites in the leaves for the phytoplasma than its influence on the feeding behavior and insect benefit in this section.

It is evenly important to discuss what metabolites were downregulated as they may play a crucial role in the plant defense against insect feeding.

“The phytoplasma infected Nerium indicum showed a lower amount of Proline, Methionine, Valine, Isoleucine, Asparagine, Aspartic acid, Histidine, Tryptophan, Threonine, and Phenylalanine than healthy N. indicum [52]”: N. indicum was not infected by the phytoplasmas in that study.

“Higher amounts of low-molecular carbohydrates, free amino acids, and free fatty acids in JWB-infected leaves could explain the feeding preference of H. hamatus.” – It is not a conclusion but an assumption.

5. Conclusions

“Higher amounts of low-molecular carbohydrates, free amino acids, and free fatty acids in JWB-infected leaves could explain the feeding preference of H. hamatus.” – It is an assumption not a conclusion.

I would recommend basing your conclusions on what influence the phytoplasma exerted on the plant metabolite composition of the leaf and the insect feeding behavior.

Author Response

Thanks very much for the constructive comments and suggestions for this paper.

  • The grammar and style mistakes were corrected by a native English speaking specialist.
  • The insect species authorship for all first mentions of insect species was added as you pointed out.
  • Whether jujube leaves and the leafhopper individuals carried JWB phytoplasma was determined by amplifying 16S-rDNA fragments of JWB phytoplasma prior to EPG experiments.
  • The term “active ingestion” is the same as the term “active xylem ingestion”. They are now unified as “active xylem ingestion”.

Additionally, according to the table 1. data the insects fed significantly more time on the xylem (I presume that the term active ingestion corresponds to active xylem ingestion) than on the phloem. Why the phloem feeder fed for so long on the xylem? It can raise a discussion that the change of the xylem metabolites affected the feeding behavior.

  • This suggestion is quite constructive. But we did not separate phloem and xylem from the jujube plants. The change in metabolic composition of phloem and xylem between JWB-infected and healthy plants was not analyzed.

More emphasis should be put on the fact that the metabolites of the whole leaf were analyzed in this study. This only partially and indirectly can suggest that the arrangement of metabolites in the phloem also changed, and the data do not show the actual composition of the phloem sap in this study. The statement that only the increase of seldom metabolites changed the feeding behavior is inaccurate because the opposite can also be true. I recommend revising the manuscript sections containing such statements and rewrite them accordingly.

  • Thank you very much. We rewrote these sections containing such statements in the manuscript.

Title

The title was changed as “‘Candidatus Phytoplasma ziziphi’ changes the metabolites composition of jujube tree leaves and affects the feeding behavior of its insect vector Hishimonus hamatus Kuoh”

Simple summary/abstract

“We found that more small-molecular carbohydrates, free amino acids, and free fatty acids were accumulated in JWB-infected leaves, which attracted more vector leafhopper feeding with a higher frequency.”

In my opinion, the observed higher amounts of some metabolites in the whole leaf can only partially prove that this could be the cause of the change in the feeding behavior. The actual change of the phloem metabolites is not reported in this study. It can also be argued that the decrease of some metabolites partaking in the plant defense could produce the same effect. Additionally, the metabolomics data only showed the general trend in the whole leaf. They did not show if the same trend is present in the phloem sap, and what change it inflicted on the compounds in the phloem.

Reply: Thank you very much. “The metabolites in the phloem” were corrected as “the metabolites in the whole leaves”.

  1. Introduction

I recommend including introductory information regarding electropenetrography and its use in the phytoplasma insect-vector research.

Reply: Thanks a lot. We have added references related to the use of EPG in phytoplasma insect-vector interaction in brief.

  1. Materials and Methods

The sticky trap method can only prove the presence of the insects in the orchard. It does not prove phytoplasma infection.

Reply: Yes. This sentence was removed.

There is no information in this section on what method was used to check for the phytoplasma infection in the plants used for the insect rearing, noninfected/infected plants, and insects. The insects were caught in the orchard so there is a higher probability that some of them could have been infected.

Reply: We collected Hishimonus hamatus from jujube orchards where almost all the jujube leaves appeared to have witches’ bloom symptoms. We randomly selected 100 individuals and extracted their DNA. Then we amplified a segment of 16S ribosomal DNA (rDNA) of JWB phytoplasma using PCR. The results of agarose gel electrophoresis showed that 82% of the individuals carried JWB phytoplasma (Chen et al. 2023).

Chen Y.X.; Liu R.C.; Liu J.C.; Chen X.L.; Li B.L.; Li G.W. Species identification, population dynamics and screening of leafhoppers as potential vectors of the jujube witches'-broom in jujube orchards in Yanchuan, Shaanxi province. Chinese Journal of Applied Entomology 2023, 60: 980-989, doi: 10.7679/j.issn.20951353.2023.091

Also, there is no information how the infected plants were acquired, and how it was confirmed that the plants were infected with `Ca. Phytoplasma ziziphi’.

Reply: Healthy jujube plants were two-year old seedlings away from any insect feeding. The JWB-infected status was determined by both witches’-broom symptoms and 16S rDNA detections.

  • Electropenetrography

It is reported that Hishimonus sellatus Uhler species has diurnal changes in stylet insertion behavior. Does H. hamatus possess similar behavior? Did you adjust your experiment regarding this fact? Please comment.

Reply: Sorry, we did not consider the diurnal changes in H. hamatus feeding behavior. This is an interesting phenomenon. We will observe the diurnal feeding behavior of H. hamatus in the future. 

  • EPG Statistic Analysis

In this section it should be clearly stated that all statistical analyses were performed using the R based software or it should be pointed out what other software was used in each case.

Reply: Thanks. We have added this statement to the manuscript as you pointed out. All statistical analyses were performed using the R based software.

  1. Results

3.1. Stylet probing behavior of H. hamatus

The feeding behavior describing terms are not uniform throughout this and Discussion paragraphs which makes it difficult to read and analyze the manuscript. I suggest unifying the terms according to the Table 1.

Reply: Thanks. The feeding behavior describing terms in results and discussion paragraphs were unified according to Table 1.

Please clarify if the term total duration of active ingestion is the same as the term active xylem ingestion. If yes, please use active xylem ingestion in the Table 1.

Reply: Yes. “active ingestion” and “active xylem ingestion” are same. We used the term “active xylem ingestion”.

Please clarify if the terms phloem salivation and phloem phase are the same as the phloem ingestion in Table 1.

Reply: “Phloem salivation” and “phloem phase” are the same.

Please could you explain why the median values are used in the table? It is known that medians can ignore extreme values, and in that way can skew the view on the distribution of the values.

Reply: Because the mean value is easy to suffer from extreme values. The distribution of counts, durations, and percentages of probing time for each feeding status is not normal. Therefore, we considered median values to be appropriate to describe the population distribution of each variable in Table 1.

I suggest providing a supplementary table for the data and values used in the text of this section as they are not included in other tables, and their origin is not explained

Reply: Thank. We provided the a supplementary table for the data and values of EPG.

  1. Discussion

I suggest moving the tables and figures to the results paragraph.

Reply: Thanks. Table 3 was moved to the results paragraph. Figure 3-5 were left in the discussion, because the changes in each individual compound were not described in the results section.

The phytoplasmas reside in the phloem, and the insects are also phloem feeders. This fact requires that more focus should be given to the composition of the metabolites in this tissue.

Reply: Thanks. As you pointed out, the accumulated levels of metabolites from phloem could provide more immediate evidence to explain the feeding preference of the vector leafhopper. But we did not separate the phloem from the whole leaf. We will detect the phloem metabolic composition in the future.

It was more discussed what benefit was brought by the increase of some metabolites in the leaves for the phytoplasma than its influence on the feeding behavior and insect benefit in this section.

It is evenly important to discuss what metabolites were downregulated as they may play a crucial role in the plant defense against insect feeding.

Reply: Thanks very much. We discussed the down-accumulation of lignans in JWB-infected leaves compared to healthy leaves.

“The phytoplasma infected Nerium indicum showed a lower amount of Proline, Methionine, Valine, Isoleucine, Asparagine, Aspartic acid, Histidine, Tryptophan, Threonine, and Phenylalanine than healthy N. indicum [52]”: N. indicum was not infected by the phytoplasmas in that study.

Reply: Sorry. That is an error. The description of this reference was removed.

“Higher amounts of low-molecular carbohydrates, free amino acids, and free fatty acids in JWB-infected leaves could explain the feeding preference of H. hamatus.” – It is not a conclusion but an assumption.

Reply: Thanks. It should not be an conclusion.

  1. Conclusions

“Higher amounts of low-molecular carbohydrates, free amino acids, and free fatty acids in JWB-infected leaves could explain the feeding preference of H. hamatus.” – It is an assumption not a conclusion.

I would recommend basing your conclusions on what influence the phytoplasma exerted on the plant metabolite composition of the leaf and the insect feeding behavior.

Reply: Thanks for your suggestion. The conclusion is rewrite as follows:

The results of this work lead us to the conclusion that the leafhopper H. hamatus exhibited distinct feeding behavior when consuming JWB-infected jujube leaves com-pared to healthy leaves. Specifically, the leafhoppers that fed on JWB-infected leaves displayed increased frequency and a longer duration of phloem ingestions. This altered feeding behavior might attribute, in part, to the differing chemical composition be-tween the two types of leaves. This research could contribute to a deeper understanding of plant-phytoplasma-insect interactions and their implications for the spread of phytoplasma-associated diseases.

Reviewer 2 Report

In this paper, the effects of Candidatus Phytoplasma ziziphi’ infections on feeding behavior of its natural vector, the leafhopper Hishimonus hamatus, and on chemical composition of jujube witches’-broom (JWB)-affected jujube plants were investigated through electrical penetration graph (EPG) technique and metabolomic analysis, respectively. Data obtained showed that H. hamatus had more phloem ingestions and a longer feeding activity on JWB-infected jujube plants than on healthy plants. This behaviour could be related to the higher amounts of low-molecular carbohydrates, free amino acids, and free fatty acids occurring in diseased plants. The information provided are relevant to phytoplasma research community. Also, the work was properly conducted and the paper is well organized. However, the following points need to be corrected or improved before acceptance:

Lines 11-12. The sentence “Failure of phytoplasma artificial cultivation makes the research on phytoplasma lag other plant diseases” is not clear and should be redrafted.

Lines 13-15. More properly as: “In this study, the feeding behavior of the leafhopper Hishimonus hamatus fed on healthy Chinese jujube leaves and on jujube witches’-broom (JWB)-affected leaves was investigated to clarify whether H. humatus prefers diseased leaves”.

Lines 82-84, and 143. Authors should provide references and/or clear evidence that JWB-diseased plants had previously been examined for phytoplasma infection and the identity of the detected phytoplasma was established.

Lines 84-85. “The infection was confirmed by yellow sticky card trapping” Do you mean infestation? Please clarify.

English could be improved.

Author Response

Thanks very much for your suggestions. The corrections are listed as follows:

Lines 11-12. The sentence “Failure of phytoplasma artificial cultivation makes the research on phytoplasma lag other plant diseases” is not clear and should be redrafted.

Reply: Thanks. We changed this sentence as “The difficulty of phytoplasma artificial cultivation makes research on the interactions of phyto-plasma-plant-vector insects lag behind the interactions of other phytopathogen-plant-vector in-sects.”

Lines 13-15. More properly as: “In this study, the feeding behavior of the leafhopper Hishimonus hamatus fed on healthy Chinese jujube leaves and on jujube witches’-broom (JWB)-affected leaves was investigated to clarify whether H. humatus prefers diseased leaves”. 

Reply: Thanks. We corrected this sentence as you suggested.

Lines 82-84, and 143. Authors should provide references and/or clear evidence that JWB-diseased plants had previously been examined for phytoplasma infection and the identity of the detected phytoplasma was established.

Reply: Thanks. We added the examination methods as following: Prior to the initiation of EPG experiments, a preliminary assessment was conducted to determine whether the tested leafhopper carried JWB phytoplasma. This assessment involved amplifying a segment of 16S ribosomal DNA (rDNA) of JWB phytoplasma using a polymerase chain reaction (PCR) technique as described by Jung et al. [36].

Reference 36:  Jung, H.Y.; Sawayanagi, T.; Kakizawa, S.; Nishigawa, H.; Wei, W.; Oshima, K.; Miyata, S.I.; Ugaki, M.; Hibi, T.; Namba, S. ‘Candidatus Phytoplasma ziziphi’, a novel phytoplasma taxon associated with jujube witches'-broom disease. Int. J. Syst. Evol. Micr. 2003, 53, 1037-1041, doi:10.1099/ijs.0.02393-0.

Lines 84-85. “The infection was confirmed by yellow sticky card trapping” Do you mean infestation? Please clarify.

Reply: This is an error. This sentence was removed.

Comments on the Quality of English Language could be improved.

Reply: Thanks. The grammar and spelling mistakes were corrected by a native English speaker specialist.

Round 2

Reviewer 1 Report

Line 13. lag behind the interactions of other phytopathogen-plant-vector insects.    This part needs to be clarified. I suggest: “lag behind the research on interactions ……”

Line 16: clarify whether H. humatus prefers diseased leaves. - The methods in the manuscript do not allow to clarify this. These methods can show if the infection changed the feeding behavior or feeding preference.

humatus – hamatus

Line 17: access – I suggest: survey, research, inspect.

Chemical components – I suggest using metabolite composition term as it would be more precise, and it corresponds better to the methods and the data in your study. It is advisable to choose one term and use it throughout the manuscript. The use of different terms with the same meaning can confuse the reader.

Line 22: that feeds on.

Line 23: This leafhopper species has been verified to transmit jujube witches’ broom (JWB), a fatal phytoplasma plant pathogen, which belongs to the phytoplasma subgroup 16SrV-B.

I suggest:

…jujube witches’ broom (JWB)disease, caused by the phytoplasma: a fatal plant pathogen which belongs to .......

Line 28: when feeding jujube leaves compared to healthy leaves using the Electrical Penetration Graph (EPG) technique. Feeding on healthy and infected leaves was compared?

Line 29: meta-targeted – in the manuscript widely targeted metabolome analysis is used.

Line 33: phloem phase of feeding - phloem feeding phase.

Line 36: cormarins - coumarins

Line 44: Phytoplasmas

to class Mollicutes

Line 45: inhabits in the – inhabit the

Line 46: responding – responsible.

Line 49: to reprogram – which affect. 

Line 51: fruit tree

Line 53: the side area – the region

Line 56: that – to

Line58-59: However, the jujube in China has been suffering from jujube witches’ brooms (JWB), a phytoplasma plant disease, showing symptoms including …

I suggest: However, the jujube in China has been suffering from jujube witches’ broom (JWB) disease, showing symptoms like….

Line 61: ‘Candidatus phytoplasma ziziphi’ - ‘Candidatus Phytoplasma ziziphi’

Line 62: 16SrV phytoplasma group

Phytoplasma group or phytoplasma subgroup should be added to the 16Sr~ always. It makes the sentence precise. 16Sr without explanation can mean anything.

elm yellows

Line 63: representative – not needed

Line 63-64: It is not clear what are SJP1-3 and Zaofeng6. Please clarify.

Line 67: Phytoplasmas are – you are writing about multiple instances

Line 69: flavescence dorée of grapevine (FD), caused by FD phytoplasma that belongs to the 16SrV-C or 16SrV-D subgroup, - I suggest: flavescence dorée (FD) disease of grapevine, caused by FD phytoplasma that belongs to the 16SrV-C or 16SrV-D phytoplasma subgroups

Line 71: Ca. P. tritici - ` and ‘ signs are missing. Please check all the instances of `Candidatus Phytoplasma ~' for the missing signs and nonuniform capitalization in the manuscript.

16SrI - 16SrI phytoplasma group.

Line 72: researchers- researches

Line 83: and has been employed to analyze the behavior of phytoplasma vector insects – I suggest deleting or revising as it makes no sense in this sentence.

Line 90: Gas Chromatography-Mass Spectrometer (GC-MS) – this contradicts the method used in the methods section. You have used Ultra-high performance liquid chromatography (UPLC-MS/MS).

Materials and methods section still lacks the information on how the infected jujube plants were obtained.

Line 121: consistant – consistent

Line 190, 192: phloem salivation term is used but it is not explained that it corresponds to phloem ingestion and duration of phloem ingestions terms in the table 1. Please clarify or unify the term usage.

Table 1: a, b and ab markings are not explained in the footnotes.

Total duration of active xylem ingestion – [min] is missing

Line 245: Phytoplasma – phytoplasma. Capitalization is not needed.

Line 292-293: as phytoplasma have not been cultivated in vitro, relatively limited interactions of phytoplasma insect vector interactions were determined compared to other interactions of phytopathogen-insect. – It is hard to understand to what kind of interactions the authors are referring to. Please revise or delete.

Line 301: S. titanus – I suggest providing full name of the species.

Line 302: a severe grapevine disease that is infected by 16SrV-C or 16SrV-D phytoplasma subgroup – please revise or delete as this part makes no sense.

Line 308: Ca. Phytoplasma mali - ` and ‘ signs are missing

Line 313: further – more

Line 314: chemical composition – metabolite composition would be more precise term as the performed analyses and the data in the manuscript all refer to metabolite changes.

Line 332: Sedoheptuiose - Sedoheptulose

Line 351-352: lipid assembly their host plants - lipid assembly from their host plants.

Line 369: was found in JWB-infected jujube leaves were observed in JWB-infected jujube leaves – please revise.

Line 372: also – it is not needed there. Please revise.

16SrV-B subgroup - 16SrV-B phytoplasma subgroup

4.2.1 Changes of carbohydrates, 4.2.2 Changes of lipids – In these sections, only the possible impact on phytoplasma was discussed. Please could you clarify why their influence on insect feeding behavior was not discussed?

4.2.4 Changes of lignans – In the abstract and discussion sections the coumarins were mentioned. Please could you clarify why their role was not discussed in this section?

Additionally, from table 3 we can see that the amount of terpenoids decreased significantly. Please could you clarify why this result was not discussed? 

Line 399: preference for these leaves. – I suggest: preference to feed on the infected leaves.

Author Response

Thank you sincerely for your valuable and constructive suggestions. We have added a discussion regarding the relationship between carbohydrates, lipids, coumarins, and triterpenoids and the feeding preference of H. hamatus on JWB-infected and healthy jujube leaves. Additionally, we have included methods about the cultivation and detection of JWB phytoplasma infested status of healthy and JWB-infected jujube plants. Other corrections were listed below as you suggested

Comments and Suggestions for Authors

Line 13. lag behind the interactions of other phytopathogen-plant-vector insects.    This part needs to be clarified. I suggest: “lag behind the research on interactions ……”

Line 16: clarify whether H. humatus prefers diseased leaves. - The methods in the manuscript do not allow to clarify this. These methods can show if the infection changed the feeding behavior or feeding preference.

Reply: We have included methods about the cultivation and detection of JWB phytoplasma infested status of healthy and JWB-infected jujube plants in line 111 to 118.

humatus – hamatus

Line 17: access – I suggest: survey, research, inspect.

Reply: We used “inspect”

Chemical components – I suggest using metabolite composition term as it would be more precise, and it corresponds better to the methods and the data in your study. It is advisable to choose one term and use it throughout the manuscript. The use of different terms with the same meaning can confuse the reader.

Reply: Thank you. The term “metabolite composition” was unified.

Line 22: that feeds on.

Reply: Hishimonus hamatus Kuoh is a leafhopper species native to China that feeds on Chinese jujube leaves.

Line 23: This leafhopper species has been verified to transmit jujube witches’ broom (JWB), a fatal phytoplasma plant pathogen, which belongs to the phytoplasma subgroup 16SrV-B.

I suggest:

…jujube witches’ broom (JWB)disease, caused by the phytoplasma: a fatal plant pathogen which belongs to .......

Reply: corrected as “This leafhopper species has been verified to transmit jujube witches’ broom (JWB) disease, caused by phytoplasma: a fatal plant pathogen which belongs to the phytoplasma subgroup 16SrV-B.”

Line 28: when feeding jujube leaves compared to healthy leaves using the Electrical Penetration Graph (EPG) technique. Feeding on healthy and infected leaves was compared?

Reply: corrected as “To address this, a study was conducted to compare the feeding patterns of H. hamatus when feeding JWB-infested jujube leaves to healthy leaves using the Electrical Penetration Graph (EPG) technique.”

 Line 29: meta-targeted – in the manuscript widely targeted metabolome analysis is used.

 Reply: This term was unified as “widely targeted metabolome” as you advised.

Line 33: phloem phase of feeding - phloem feeding phase.

 Line 36: cormarins - coumarins

 Reply: Sorry, “cormarins” is an error.

Line 44: Phytoplasmas

 Reply: Yes, it should be a plural noun.

to class Mollicutes

 Reply: This sentence is correct as “Phytoplasma to Mollicutes are prokaryotic plant pathogens, sharing similar morphology and ultrastructure with mycoplasma” as you suggested.

Line 45: inhabits in the – inhabit the

 Reply: Thank you. “Inhabit” is a transitive verb.

Line 46: responding – responsible.

Line 49: to reprogram – which affect.

Line 51: fruit tree

Line 53: the side area – the region

 Reply: Because only the Yellow River flows through the Shanxi-Shaanxi Gorge, and there is no land in this gorge. The jujube trees grow on the side of this gorge.

Line 56: that – to

 Reply: Given the rich nutritional and medical value of jujube fruits and the adaptivity of ju-jube trees to tolerate arid and barren soil [10–13], the jujube cultivating area spreads worldwide [8,10].

Line58-59: However, the jujube in China has been suffering from jujube witches’ brooms (JWB), a phytoplasma plant disease, showing symptoms including …

I suggest: However, the jujube in China has been suffering from jujube witches’ broom (JWB) disease, showing symptoms like….

 Reply: However, the jujube in China has been suffering from jujube witches’ brooms (JWB) disease, showing symptoms like witches’ broom, phyllody, leaf yellowing, and weak trees [14].

Line 61: ‘Candidatus phytoplasma ziziphi’ - ‘Candidatus Phytoplasma ziziphi’

Line 62: 16SrV phytoplasma group

Phytoplasma group or phytoplasma subgroup should be added to the 16Sr~ always. It makes the sentence precise. 16Sr without explanation can mean anything.

elm yellows

 Reply: Thank you. This sentence was corrected as “JWB was caused by the infection of JWB phytoplama ‘Candidatus Phytoplasma ziziphi’, which belongs to the 16 SrV group (the elm yellow group), 16SrV-B subgroup [16].”

Line 63: representative – not needed

 Reply: “Representative” is removed.

Line 63-64: It is not clear what are SJP1-3 and Zaofeng6. Please clarify.

 Reply: We added “JWB phytoplasma effector” before “SJP1-3” and “Zaofeng6”, respectively.

Line 67: Phytoplasmas are – you are writing about multiple instances

 Reply: Yes. “Phytoplasma” should be corrected as “Phytoplasmas”

Line 69: flavescence dorée of grapevine (FD), caused by FD phytoplasma that belongs to the 16SrV-C or 16SrV-D subgroup, - I suggest: flavescence dorée (FD) disease of grapevine, caused by FD phytoplasma that belongs to the 16SrV-C or 16SrV-D phytoplasma subgroups

 Reply: Thank you. Your description is more clear.

Line 71: Ca. P. tritici - ` and ‘ signs are missing. Please check all the instances of `Candidatus Phytoplasma ~' for the missing signs and nonuniform capitalization in the manuscript.

 Reply: Thanks. All the single quotes out of the scientific names of phytoplsamas were added.

16SrI - 16SrI phytoplasma group.

Line 72: researchers- researches

Line 83: and has been employed to analyze the behavior of phytoplasma vector insects – I suggest deleting or revising as it makes no sense in this sentence.

 Reply: “and has been employed to analyze the behavior of phytoplasma vector insects” was deleted as you advised.

Line 90: Gas Chromatography-Mass Spectrometer (GC-MS) – this contradicts the method used in the methods section. You have used Ultra-high performance liquid chromatography (UPLC-MS/MS).

 Reply: That is an error. “UPLC-MS/MS” is correct.

Materials and methods section still lacks the information on how the infected jujube plants were obtained.

 Reply: We added this information.

Line 121: consistant – consistent

Line 190, 192: phloem salivation term is used but it is not explained that it corresponds to phloem ingestion and duration of phloem ingestions terms in the table 1. Please clarify or unify the term usage.

 Reply: We unified this term as “phloem ingestion”

Table 1: a, b and ab markings are not explained in the footnotes.

Total duration of active xylem ingestion – [min] is missing

 Reply: We added “Different letters in each column represented that there is significant difference among different groups after post-hoc comparisons” in the footnotes of Table 1. “[min]” is added.

Line 245: Phytoplasma – phytoplasma. Capitalization is not needed.

 Line 292-293: as phytoplasma have not been cultivated in vitro, relatively limited interactions of phytoplasma insect vector interactions were determined compared to other interactions of phytopathogen-insect. – It is hard to understand to what kind of interactions the authors are referring to. Please revise or delete.

 Reply: “However, as phytoplasma have not been cultivated in vitro, relatively limited interactions of phytoplasma-insect vector interactions were determined compared to the study of other interactions of phytopathogen-insect. This led to relatively few research using EPG on phytoplasma-insect-plant interactions.”

This sentence is deleted.

Line 301: S. titanus – I suggest providing full name of the species.

 Reply: We added the full name: Scaphoideus titanus Ball

Line 302: a severe grapevine disease that is infected by 16SrV-C or 16SrV-D phytoplasma subgroup – please revise or delete as this part makes no sense.

Reply: “a severe grapevine disease that is infected by 16SrV-C or 16SrV-D phytoplasma subgroup,” was removed.

Line 308: Ca. Phytoplasma mali - ` and ‘ signs are missing

 Reply: the missing quotes were added.

Line 313: further – more

 Reply: This sentence was corrected as “To gain a deeper understanding of these dynamics, more EPG data focused on phytoplasma-insect vectors is necessary.”

Line 314: chemical composition – metabolite composition would be more precise term as the performed analyses and the data in the manuscript all refer to metabolite changes.

 Reply: Thanks. We replaced term “chemical composition” with “metabolite composition”

Line 332: Sedoheptuiose - Sedoheptulose

 Reply: Thanks. “Sedoheptuiose” is an error.

Line 351-352: lipid assembly their host plants - lipid assembly from their host plants.

 Reply: Thanks.

Line 369: was found in JWB-infected jujube leaves were observed in JWB-infected jujube leaves – please revise.

 Reply: I am sorry. “was found in JWB-infected jujube leaves were observed in JWB-infected jujube leaves” should be “were observed in JWB-infected jujube leaves”.

Line 372: also – it is not needed there. Please revise.

 Reply: “also” was removed.

16SrV-B subgroup - 16SrV-B phytoplasma subgroup

 Reply: Thanks. “16SrV-B subgroup” is not clear as you pointed out.

4.2.1 Changes of carbohydrates, 4.2.2 Changes of lipids – In these sections, only the possible impact on phytoplasma was discussed. Please could you clarify why their influence on insect feeding behavior was not discussed?

4.2.4 Changes of lignans – In the abstract and discussion sections the coumarins were mentioned. Please could you clarify why their role was not discussed in this section?

Additionally, from table 3 we can see that the amount of terpenoids decreased significantly. Please could you clarify why this result was not discussed?

 Reply: Thanks a lot. We have added some discussion regarding the relationship between carbohydrates, lipids, coumarins, and triterpenoids, and the feeding preference of H. hamatus on JWB-infected and healthy jujube leaves.

Line 399: preference for these leaves. – I suggest: preference to feed on the infected leaves.

 Reply: Thanks.